# Optimal Survey Design for Private Mean Estimation

Yu-Wei Chen [1]   Raghu Pasupathy [1]   Jordan Awan [1]

## Abstract

This work identifies the first privacy-aware stratified sampling scheme that minimizes the variance for general private mean estimation under the Laplace, Discrete Laplace (DLap) and Truncated-Uniform-Laplace (TuLap) mechanisms within the framework of differential privacy (DP). We view stratified sampling as a subsampling operation, which amplifies the privacy guarantee; however, to have the same final privacy guarantee for each group, different nominal privacy budgets need to be used depending on the subsampling rate. Ignoring the effect of DP, traditional stratified sampling strategies risk significant variance inflation. We phrase our optimal survey design as an optimization problem, where we determine the optimal subsampling sizes for each group with the goal of minimizing the variance of the resulting estimator. We establish strong convexity of the variance objective, propose an efficient algorithm to identify the optimal design, and offer insights on the structure of this design in different settings.

## 1. Introduction

Differential Privacy (DP), introduced by Dwork et al. (2006), is a popular probabilistic framework designed to protect individual privacy while preserving the utility of data. By introducing calibrated random noise into the data processing, DP ensures that outputs remain informative while reducing the risk of identifying individuals. However, this added noise introduces unique challenges for data analysis. Neglecting the effects of DP mechanisms can lead to biased and incorrect conclusions (Santos-Lozada et al., 2020; Kenny et al., 2021). To address these challenges, researchers commonly employ various inference strategies, such as Bayesian inference (Bernstein & Sheldon, 2018 & 2019; Schein et al., 2019; Kulkarni et al., 2021; Ju et al., 2022), asymptotic anal-

ysis (Gaboardi et al., 2016; Gaboardi & Rogers, 2017; Wang et al., 2018), simulation-based inference (Awan & Wang, 2024), and bootstrapping methods (Ferrando et al., 2022; Wang et al., 2022). However, there is also a growing need to integrate DP into the design of data collection schemes.

Survey sampling traditionally encompasses three components: sample selection, data collection, and estimation (Brick, 2011). Over time, survey sampling has been evolving to incorporate new technologies (Frankel & Frankel, 1987), such as registration-based sampling (Green & Gerber, 2006), telephone sampling (Force et al., 2010), and computerization (Baker, 1998). Differential privacy represents one of the latest advances, fostering the need to optimize data collection to balance privacy and utility.

Among survey sampling methods, stratified sampling stands out as a robust scheme that leverages auxiliary information to collect valuable samples that can minimize variance. Unlike simple random sampling (SRS), stratified sampling minimizes the risk of having bad samples by dividing the population into groups (strata) based on common characteristics (Lohr, 2021). Neyman (1934) was the first to formalize stratified sampling, introducing an optimal allocation of samples to minimize variance across groups—a goal aligned with the principles of experimental design (Wu & Hamada, 2011).

Developing DP techniques for surveys is very important to protect individual respondents, especially when sensitive questions are asked. Furthermore, another key motivation for incorporating DP is as a technique to reduce response bias—also known as answer bias—which often arises when individuals avoid answering sensitive or controversial questions truthfully, leading to skewed or inaccurate conclusions. Randomized response, introduced by Warner (1965), provides a mechanism for respondents to address such questions while satisfying differential privacy (Dwork et al., 2014). Incorporating appropriate noise through DP techniques, our framework effectively balances data utility with individual privacy and can also reduce response biases.

When considering differential privacy for survey sampling, it is first important to recognize a crucial result of differential privacy under sampling which plays a key role in the formulation of our problem: When a privacy mechanism is applied to a randomly sampled subset of a population (while the

[1]Department of Statistics, Purdue University, West Lafayette IN, USA. Correspondence to: Jordan Awan <jawan@purdue.edu>.

*Proceedings of the 42nd International Conference on Machine Learning*, Vancouver, Canada. PMLR 267, 2025. Copyright 2025 by the author(s).

sampled individuals themselves remain secret), a stronger privacy guarantee can be achieved (Kasiviswanathan et al., 2011). This effect is referred to as the "secrecy of the sample" or privacy amplification by subsampling. Thus, in stratified sampling, where the population is divided into subpopulations, subsampling within groups can amplify privacy protection. This effect adds complexity to the optimization problem of determining the optimal survey design when integrating differential privacy into stratified sampling.

This paper is the first to consider optimizing a survey design when incorporating a differential privacy guarantee. Specifically, we develop an optimal stratified sampling scheme to minimize the estimator variance in private mean estimation under differential privacy. Holding the total sample size fixed, we search for the optimal subgroup sizes. A key challenge is that different subsampling rates for each group require different "nominal privacy budgets" in order to give the same privacy guarantee to all members of the population, which results in a complex objective function.

Ignoring DP-induced variance during the design phase can lead to significant inflation in estimator variance, as demonstrated in Section 5 and highlighted in the table below. Essentially, the Neyman allocation (Neyman, 1934), referred to as naive stratified sampling in our setting, only minimizes the variance from the data but can perform poorly when incorporating the variance induced by DP. Since the DP variance objectives in our problem contain both a component from the data and one from the DP mechanism, the DP-optimal design differs from the Neyman allocation.

Table 1 compares the non-private (Neyman) solution and the DP-optimal design, both evaluated under the DP variance objective, using Laplace and Truncated-Uniform-Laplace (TuLap) mechanisms (Awan & Slavković, 2018). The table reports the ratio of the DP variance objectives—Neyman over optimal—highlighting the potential inefficiency caused by ignoring the effect of DP during the design.

| $\epsilon$ | 0.1 | $10^{-1/2}$ | 1 | $10^{1/2}$ | 10 |
|---|---|---|---|---|---|
| Laplace | 1.828 | 2.095 | 2.269 | 2.311 | 1.973 |
| TuLap | 2.405 | 3.324 | 3.877 | 4.060 | 4.076 |

*Table 1.* Ratio of the DP variance objective when using the non-private solution versus the DP-optimal design for population mean estimation (1 is optimal, higher is worse).

**Contributions:** We propose a novel framework for designing stratified sampling schemes under a hybrid local/central differential privacy regime, leveraging a design of experiment (DOE) perspective. We provide an efficient algorithm for locating the optimal integer design, which practitioners can easily implement in practice. Our approach accounts

for variance from sampling as well as from the DP noise amplified by subsampling during the design phase, allocating the best subsampling sizes to minimize the variance of our final estimator.

This work is, to the best of our knowledge, the first to apply experimental design principles to data collection under differential privacy, fundamentally altering optimal stratified sampling schemes to accommodate DP considerations. Our contributions include the following:

- We identify and formulate the problem as a constrained integer-programming problem, identifying its alignment within the framework of DOE.
- We establish strong convexity for a general variance objective, covering important cases such as A-optimality (minimizing the trace of the covariance matrix) and population mean estimation, under three common additive DP mechanisms: Laplace, Discrete Laplace, and Truncated-Uniform-Laplace.
- For the population mean estimate, we derive closed-form continuous solutions under Discrete Laplace and Truncated-Uniform-Laplace mechanisms; additionally, we derive the optimal continuous design when using purely Laplace noise, which reveals the DP-aware design lies between the original, no-noise design and the pure Laplace noise design.
- By leveraging the strong convexity of the variance objectives, we develop a computationally efficient algorithm to locate the integer-optimal design, overcoming the intractability of exhaustive search methods.

**Organization:** The rest of the paper is structured as follows: Section 2 provides the necessary background on local and central differential privacy as well as privacy amplification by subsampling. Section 3 formulates the main problem, a convex-constrained minimization problem with a general variance objective. Section 4 establishes the strong convexity of the problem, a key property enabling the efficient search for the integer-optimal design. In Section 5, we illustrate variance inflation resulting from naive stratified sampling without considering DP effects and demonstrate the efficiency of our algorithm in locating the integer-optimal design, even as the number of groups increases. Finally, Section 6 discusses the implications of our findings and potential avenues for future work.

**Related Work:** Although optimal survey design for private estimation remains unexplored, differentially private survey sampling has recently been studied in other contexts. Lin et al. (2024) construct confidence intervals for proportions using data collected through stratified sampling. Bun et al. (2020) examine stratified and cluster sampling, highlighting that certain sampling schemes can degrade privacy rather than enhance it and can increase privacy risks.

Sampling has also been employed as a technique to address DP-related problems. Ebadi et al. (2016) examine the impact of various sampling schemes on differential privacy, demonstrating that only Bernoulli sampling amplifies privacy protection. Joy & Gerla (2017) propose a sampling-based privacy mechanism satisfying differential privacy, while Bichsel et al. (2018) develop a correlated sampling method to detect privacy violation.

The concept of "secrecy of the sample," proposed and formalized by Kasiviswanathan et al. (2011), highlights the role of random sampling from population in enhancing privacy guarantees while keeping the members of the dataset secret. Li et al. (2012) demonstrate that implementing $k$-anonymity safely after a random sampling step ensures $(\epsilon, \delta)$-DP. Cheu et al. (2019) employ "secrecy of the sample" to establish the privacy guarantee of the shuffled model, an intermediate variant between central and local models that enhances privacy by relying on a trusted curator to shuffle the locally privatized data before releasing a final private statistic. Arcolezi et al. (2021) incorporate random sampling into their solution for multivariate frequency estimation in locally differentially private (LDP) settings.

Variance minimization and estimation, as well as utility-maximization mechanisms, have been widely investigated under DP. Treating multi-agent systems as probabilistic models of environmental states parameterized by agent profiles, Wang et al. (2017) establish a lower bound on the $l_1$-induced norm of the covariance matrix for minimum-variance unbiased estimators when the agents' profiles are $\epsilon$-DP. Li et al. (2023b) expose how output poisoning attacks can manipulate and deteriorate mean and variance estimation under local DP. Amin et al. (2019) identify a bias-variance trade-off caused by clamping in DP learning and provide careful tuning on the clamping bound. For a fixed count query, Ghosh et al. (2009) show that the geometric mechanism minimizes the expected loss for virtually all possible users while satisfying the DP constraint. Similarly, for a single real-valued query function, Geng & Viswanath (2015) demonstrate that the staircase mechanism can minimize $\ell_1$ and $\ell_2$ costs under specific parameter settings.

## 2. Background

We introduce the necessary background of local and central differential privacy and relevant subsampling results.

Differential privacy can be ensured from two perspectives: local DP and central DP. Both approaches achieve privacy guarantees by employing randomized mechanisms that perturb sensitive data or statistics and produce their privatized outputs. Local DP offers stronger privacy protection by privatizing individual data, ensuring that sensitive information remains unknown to anyone, thereby shielding individu-

als from both internal and external threats. However, this comes at the cost of reduced data utility. In contrast, central DP relies on trusted data curators to collect sensitive data and subsequently release a privatized summary, protecting individual only from external adversaries.

**Definition 2.1** (Local Differential Privacy: Duchi et al., 2013). Let $\mathcal{X}$ be the set of possible contributions of an individual. A privacy mechanism $M$ provides local $\epsilon$-differential privacy, if for any two data points $x, x' \in \mathcal{X}$ and for any measurable set $S \subseteq \text{Range}(M)$,

$$\Pr[M(x) \in S] \leq e^\epsilon \Pr[M(x') \in S].$$

To safeguard individual privacy through added noise, the amount of noise must be carefully quantified, with sensitivity playing a pivotal role in this process. Greater data dispersion increases sensitivity, which in turn requires scaling up the noise in the privacy mechanism.

**Definition 2.2** (Sensitivity). Let $f : \mathcal{X} \to \mathbb{R}$ be a statistic. The sensitivity of $f$ is $\Delta f = \max_{x,x' \in \mathcal{X}} |f(x) - f(x')|$.

In Example 2.3, we introduce three DP mechanisms to which we apply our findings throughout the paper.

**Example 2.3.** The following are three common DP mechanisms. For local $\epsilon$-DP, given a real-valued statistic $f(x)$,

- the Laplace mechanism is $Z = f(x) + L$, where $L \sim \text{Lap}(0, s)$ with $s = \Delta f / \epsilon$.
- the Discrete Laplace mechanism (DLap) (Inusah & Kozubowski, 2006; Ghosh et al., 2009) for integer-valued data is $Z = f(x) + K$, where $K \sim P(K = k) = \frac{1-p}{1+p} p^{|k|}$ with $p = \exp(-\epsilon/\Delta f)$.
- the Truncated-Uniform-Laplace mechanism (TuLap) (Awan & Slavković, 2018) is $Z = f(x) + K + U$, where $K$ is the same as in DLap and $U \sim \text{Uniform}(-1/2, 1/2)$. TuLap is a canonical noise distribution (Awan & Vadhan, 2023) and it is related to the Staircase distribution (Geng & Viswanath, 2015).

While local DP offers strong protection against both external adversaries as well as the data collectors themselves, it requires a large amount of noise for privatization. For example, Duchi et al. (2013) show that local DP mechanisms have inferior asymptotic variance compared to non-private estimators. On the other hand, central DP has a trusted curator, but gives the same DP guarantee to external adversaries and allows for asymptotically negligible noise to be added (Smith, 2011; Barber & Duchi, 2014).

**Definition 2.4** (Central Differential Privacy: Dwork et al., 2006). Let $\mathcal{X}^n$ be the set of possible datasets with sample size $n$ and $d_H(\cdot, \cdot)$ be the Hamming distance on $\mathcal{X}^n \times \mathcal{X}^n$, a privacy mechanism $M$ provides (central) $\epsilon$-differential privacy, if for any two datasets $X, X' \in \mathcal{X}^n$

such that $d_H(X, X') \leq 1$, and for any measurable set $S \subseteq \text{Range}(M)$,

$$\Pr[M(X) \in S] \leq e^\epsilon \Pr[M(X') \in S].$$

Note that all local DP mechanisms are also central DP. Therefore, the following lemmas on central DP can be applied to local DP mechanisms.

**Lemma 2.5** (Parallel Composition: McSherry, 2009). *Let $M_1, M_2, \ldots, M_k$ be a set of $k$ mechanisms, where each $M_i$ satisfies $\epsilon_i$-DP. Suppose these mechanisms are applied to disjoint subsets of the dataset $\mathcal{X}^n$, denoted as $D_1, D_2, \ldots, D_k$, such that $\mathcal{X}^n = \bigcup_{i=1}^k D_i$, and where the sizes of $D_i$'s are public. Then, the combined mechanism $M = (M_i)_{i=1}^k$ satisfies $\max_i \epsilon_i$-DP.*

Lemma 2.5 states that the ultimate privacy guarantee of a set of privacy mechanisms applied to disjoint datasets only hinges on the worst among all guarantees. In stratified sampling, a set $D_i$ represents a stratum (group).

**Lemma 2.6** (Subsampling: Corollary 3, Dong et al., 2022; Ullman, 2017). *If $M$ is a privacy mechanism that satisfies $\epsilon$-DP for a dataset of size $n$, and $S_m$ is the subsampling operator that chooses a subset of size $m$ from the dataset of size $n$ uniformly at random, then the subsampled mechansim $M \circ S_m$ satisfies $\log(1 - q + q \exp(\epsilon))$-DP, where $q = m/n$.*

Lemma 2.6 shows how subsampling creates its randomness, thereby bringing about privacy amplification. It follows from Lemma 2.5 that if $N$ is decomposed into $k$ disjoint groups of size $N_i$ for $i = 1, \ldots, k$, and we want to sample subsets of size $n_i$ from each group, uniformly at random, then the privacy guarantee for a nominal $\epsilon$-DP mechanism $M$ applied to the subsamples is $\log(1 - q_{\max} + q_{\max} \exp(\epsilon))$, where $q_{\max} = \max_i \frac{n_i}{N_i}$.

Finally, we recall the post-processing property of DP: If $M : \mathcal{X}^n \to \mathcal{Y}$ is an $\epsilon$-DP mechanism and $g : \mathcal{Y} \to \mathcal{Z}$ is another mechanism, then $g \circ M : \mathcal{X}^n \to \mathcal{Z}$ satisfies $\epsilon$-DP (Dwork et al., 2014). This property allows us to construct customized estimators from the DP outputs, without compromising the privacy guarantee.

## 3. Problem Setup

In this paper, we minimize the variance of a mean estimator, which comprises data randomness and additive privacy noise centered at zero, with the following problem setup:

Suppose there are $k$ groups (strata) of people $D_i$ ($i = 1, \ldots, k$) with size $N_i$ that make up the entire population. In each group $i$, $Y_{ij}$ represents the survey response of the $j$-th individual ($j = 1, \ldots, N_i$) which has mean $\mu_i$ and variance $\sigma_i^2$ with bounded support. In a local DP setting, instead of $Y_{ij}$, $Z_{ij} = Y_{ij} + W_{ij}$ is the privatized survey response,

where $W_{ij}$ is the i.i.d. additive noise with mean 0 and finite variance $\gamma^2$ depending on $n_i$, $N_i$ and $\epsilon$. Assume that $n_i$ samples are drawn from group $D_i$ with a total sample size of $\eta \in [k, \sum_{i=1}^k N_i]$. The constrained minimization problem of interest becomes

$$\arg\min \sum_{i=1}^k \frac{\alpha_i^2}{n_i} \left[ \sigma_i^2 + \gamma^2(n_i, N_i, \epsilon) \right], \qquad (1)$$

subject to $C_n := \{n \in \mathbb{N}^k : \sum n_i = \eta \text{ and } 0 < n_i \leq N_i, \forall i\}$, where $\alpha_i$ are pre-determined weights.

This problem is classified as a nonlinear integer program. As it will be addressed later using the Lagrangian, which introduces a continuous multiplier for the equality constraint, it can be generally treated as a mixed-integer programming problem (Lee & Leyffer, 2011).

Note that $\alpha_i$, $N_i$, $\sigma_i$, ($i = 1, .., k$) and $\eta$ are assumed to be given or determined prior to subsampling. The following examples show how the $\alpha_i$ can be chosen to optimize for various variance objectives.

**Example 3.1** (Population Mean Estimation). One of the most important parameters to estimate in survey sampling is the population mean. For group $i$, the group mean $\mu_i$ can be estimated by $\hat{\mu}_i = \frac{1}{n_i} \sum_{j=1}^{n_i} Z_{ij}$, which is an unbiased estimator of $\mu_i$. The population mean $\mu$ can thus be unbiasedly estimated by $\hat{\mu} = \frac{\sum_{i=1}^k N_i \hat{\mu}_i}{\sum_{i=1}^k N_i}$. Then, $\text{Var}(\hat{\mu}) = \frac{1}{(\sum N_i)^2} \sum_{i=1}^k \frac{N_i^2}{n_i} \left[ \sigma_i^2 + \gamma^2(n_i, N_i, \epsilon) \right]$. Thus, the constrained minimization problem is

$$\arg\min \sum_{i=1}^k \frac{N_i^2}{n_i} \left[ \sigma_i^2 + \gamma^2(n_i, N_i, \epsilon) \right] \qquad (2)$$

$$\text{s.t.} \sum n_i = \eta, \; n_i \leq N_i, \; n_i \in \mathbb{N}, \quad \forall i.$$

This aligns with (1) as $\alpha_i = N_i$ for all $i$.

**Example 3.2** (A-Optimal Experimental Design). Another important case is the A-optimal experimental design, where the goal is to estimate each group mean with the A-optimal weighting. Denote the covariance matrix of $(\hat{\mu}_1, \ldots, \hat{\mu}_k)^\top$ as $E_\mu$, then the A-optimal experimental design is

$$\arg\min \text{tr}(E_\mu) = \sum_{i=1}^k \frac{1}{n_i} \left[ \sigma_i^2 + \gamma^2(n_i, N_i, \epsilon) \right] \qquad (3)$$

$$\text{s.t.} \sum n_i = \eta, \; n_i \leq N_i, \; n_i \in \mathbb{N}, \quad \forall i.$$

This alplns with (1) as $\alpha_i = 1$ for all $i$.

*Remark* 3.3. There are other interesting settings that fit into this framework. For instance, one may be interested in a unit-free optimal design by setting $\alpha_i = 1/\sigma_i$.

Now, we look into the variance component from the privacy mechanism, that is, $\gamma^2(n_i, N_i, \epsilon)$ for all $i$, where the

subsampling comes into play. Rather than using $q_{\max}$ for all $k$ groups as in Lemma 2.6, we consider $q_i = \frac{n_i}{N_i}$ such that $M_i \circ S_{n_i}$ satisfies $\epsilon$-DP, where $M_i$ is the privacy mechanism for group $i$ and $S_{n_i}$ is the subsampling operator which chooses a subset of size $n_i$ from the group $i$. This ensures every person gets the same level of central DP privacy protection, regardless of their group size.

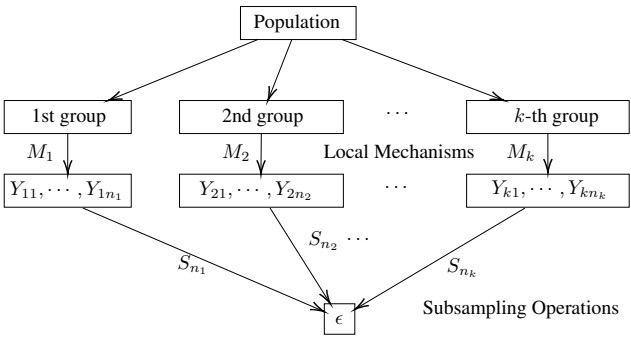

*Figure 1.* Diagram of the DP stratified sampling

Illustrated in Figure 1, the key challenge lies as follows:

- To avoid wasting the final privacy budget $\epsilon$ across groups, the parallel composition *between groups* must account for the subsampling operations when determining a central nominal budget for each group. This budget is then distributed as a local nominal budget to each individual in the group in order to protect individual survey response, ensuring—via parallel composition *within group*—that the group as a whole satisfies the central nominal privacy guarantee.

- Since the total sample size $\eta$ is fixed, allocating fewer samples to one group may allow for less local noise in that group, while simultaneously increasing the local noise required for other groups and amplifying the $1/n_i$ factor in the overall variance objective.

Therefore, our method has dual privacy guarantees: a local DP guarantee from the nominal privacy loss budget against the data collector, and a central DP guarantee, boosted by subsampling, against external adversaries.

**Proposition 3.4.** *If the nominal privacy budget of $M_i$ is $\log\left(\frac{\exp(\epsilon/\Delta f)-1+q_i}{q_i}\right)$-DP, then $M_i \circ S_{n_i}$ satisfies $\epsilon$-DP.*

Proposition 3.4 establishes the dual privacy guarantees for our mechanisms, ensuring uniform privacy protection for all individuals from public disclosure.

**Example 3.5.** Applying Proposition 3.4 to the three mechanisms with $s_i = 1/\log\left(1 + \frac{\exp(\epsilon/\Delta f)-1}{n_i}N_i\right)$, we have the following:

- The Laplace mechanism for local $\epsilon$-DP is $Z_{ij} = Y_{ij} + s_i \cdot \mathrm{Lap}(0,1)$. The variance objective is

$$\sum_{i=1}^{k} \frac{\alpha_i^2}{n_i}\left[\sigma_i^2 + 2\log^{-2}\left(1 + \frac{\exp(\epsilon/\Delta f)-1}{n_i}N_i\right)\right]. \quad (4)$$

- The Discrete Laplace mechanism for local $\epsilon$-DP is $Z_{ij} = Y_{ij} + K_{ij}$, where $K_{ij} \sim \mathrm{DLap}\left(p_i = \exp(1/s_i)\right)$. The variance objective is

$$\sum_{i=1}^{k} \frac{\alpha_i^2}{n_i}\left[\sigma_i^2 + 2\frac{\frac{n_i}{N_i}(\exp(\epsilon/\Delta f)-1+\frac{n_i}{N_i})}{(\exp(\epsilon/\Delta f)-1)^2}\right]. \quad (5)$$

- The Truncated-Uniform-Laplace mechanism for local $\epsilon$-DP is $Z_{ij} = Y_{ij} + K_{ij} + U_{ij}$ where $K_{ij} \sim \mathrm{DLap}\left(p_i = \exp(1/s_i)\right)$. and $U_{ij} \sim \mathrm{Uniform}(-\frac{1}{2}, \frac{1}{2})$. The variance objective is

$$\sum_{i=1}^{k} \frac{\alpha_i^2}{n_i}\left[\sigma_i^2 + \frac{1}{12} + 2\frac{\frac{n_i}{N_i}(\exp(\epsilon/\Delta f)-1+\frac{n_i}{N_i})}{(\exp(\epsilon/\Delta f)-1)^2}\right]. \quad (6)$$

By using an exhaustive search, the complexity of solving (1) is $O\left(\binom{\eta-1}{k-1}\right)$ (Ross, 1974), motivating the need for a customized optimization method.

## 4. Theoretical Results and Algorithm

In this section, we develop the optimal integer design which solves (1). In Section 4.1, we discover and prove the strong convexity of the variance objectives under Laplace, DLap, and TuLap mechanisms, which enables us to precisely locate the optimal design in Section 4.3. In Section 4.2 we also derive some closed-form solutions over continuous space for some special cases.

### 4.1. Strong Convexity

We begin by establishing the strong convexity of our variance objectives. Strong convexity ensures a unique optimum over the reals and provides a quadratic lower bound for the objective function. These properties are leveraged in Section 4.3 to develop an efficient optimization algorithm.

**Theorem 4.1** (Strong Convexity)**.** *Let $\alpha_i$, $N_i$, $\sigma_i$, and $\eta$ be given for all $i = 1, \ldots, k$. Then, the continuous relaxations of the variance objectives (4), (5) and (6), with $(n_1, \ldots, n_k)$ replaced by a continuous vector $x \in C_x := \{x \in \mathbb{R}^k : \sum x_i = \eta, 0 < x_i \le N_i, \forall i\}$, are strongly convex.*

*Proof Sketch.* We first prove that the variance objectives are strongly convex over $(0, \eta)^k$ and then show that this property continues to hold under the constraints. For the DLap and TuLap mechanisms, strong convexity follows by inspection; for Laplace, strong convexity is established by change of variables and repeated differentiation. $\square$

Strong convexity in Theorem 4.1 guarantees a unique solution over the reals. This solution can be found via Newton's method with established R packages such as `optim`, `nloptr` or `alabama`. However, solving the mixed-integer programming problem is more complex, as existing packages do not provide direct solutions. Strong convexity is key in identifying the integer-optimal design in Section 4.3. Note that neither CVX's free solvers in `MatLab` nor any R package support mixed-integer programming.

## 4.2. Closed-Form Solution for Population Mean

A closed-form continuous solution is desirable for its ease of implementation and the insights it provides into the behavior of the design. While such a solution does not exist for all $\alpha_i$, intriguing results emerge when $\alpha_i = N_i$, the case of population mean estimation.

**Proposition 4.2** (Closed-Form Solutions for Population Mean). *If $M_i$ is DLap or TuLap with $\alpha_i = N_i$ and assume that $[\tau_i/(\sum \tau_i N_i)]\eta \leq 1$ for all $i$, the continuous solution of (5) and (6) under the constraint of $C_x$ have a closed form $x_i^* = [(\tau_i N_i)/(\sum \tau_i N_i)]\eta$, where $\tau_i^2 = \sigma_i^2$ for DLap and $\tau_i^2 = \sigma_i^2 + \frac{1}{12}$ for TuLap.*

*Proof Sketch.* We first formulate the Lagrangian of the constrained optimization problem. The KKT conditions give a proportional relation $x_i \propto \tau_i N_i$ for all $i$. Then, the constraint provides a unique solution for the $x_i$'s. □

*Remark* 4.3. The solution of (5) is identical to the naive stratified sampling design, also known as the Neyman allocation (Neyman, 1934) or the optimal allocation (Kempf-Leonard, 2004), where the sample size allocated to each group is proportional to both the variability and the size of the group. In contrast, (6) has a regularization effect on its sample sizes that results in a different allocation.

As for the Laplace noise, although we were unable to derive a general closed-form solution for (4), we have an interesting finding in the case of population mean estimates, which offers insight in the interplay between the no-DP and purely DP solutions. First, we define the non-private variance as

$$\frac{1}{(\sum N_i)^2} \left\{ \sum_{i=1}^{k} \frac{N_i^2}{x_i} \sigma_i^2 \right\}. \tag{7}$$

Under the constraint of $C_x$, the non-private variance (7) is minimized at $x_i^* = (\sigma_i N_i / \sum \sigma_i N_i)\eta$ for all $i$. On the other hand, we can define the pure DP variance as

$$\frac{1}{(\sum N_i)^2} \left\{ \sum_{i=1}^{k} \frac{N_i^2}{x_i} \left[ 2 \log^{-2} \left( 1 + \frac{\exp(\epsilon/\Delta f) - 1}{x_i/N_i} \right) \right] \right\}. \tag{8}$$

**Proposition 4.4** (Closed-Form Solution of Purely Laplace Variance for Population Mean). *Under the constraint of $C_x$, the purely DP variance from the Laplace mechanism (8) is minimized at $x_i^* = (N_i/ \sum N_i)\eta$ for all $i$.*

*Proof Sketch.* In addition to the similar Lagrangian proof argument in Proposition 4.2, we use the result that $\varphi(y) = \frac{2}{y} \log^{-2}(1 + c/y)$ is strongly convex, as proved in Theorem 4.1, to identify the solution form. □

Proposition 4.4 indicates that the sample size allocated to each group is merely proportional to the size of the group. This corresponds to the concept of the proportional allocation (Kempf-Leonard, 2004). It is surprising that the solution is independent of $\epsilon$ as (8) can be blown up when $\epsilon$ drops. Although $\epsilon$ does not play a role in either the solution of the original variance or that of the pure DP variance, it plays a critical role in the solution of the total variance. We illustrate this phenomenon in Section 5.

## 4.3. Algorithm to Find the Optimal Integer Design

Since Theorem 4.1 only ensures the existence and uniqueness of the continuous-optimal design, in this section we use the strong convexity property to derive a small region that is guaranteed to contain the integer-optimal design. This result is leveraged in Algorithm 1 to efficiently find the integer-optimal design.

The search over integer points satisfying the convex constraint is finite, as the set of feasible integer-valued solutions, $D = \{n \in \mathbb{N}^k : \sum n_i = \eta, 0 < n_i < N_i, \forall i\}$, is inherently limited. Moreover, the strong convexity of the variance objective provides a quadratic lower bound, ensuring that a certain level set of this bound must contain the integer-optimal point. This level set can be characterized in terms of Euclidean distance, with the radius determined by the smallest eigenvalue of the objective function.

*Lemma* 4.5 (Range to Search for Integer-Optimal Design). Let $g_1, g_2, g_3$ be the objective function from (4), (5), (6) respectively, and $C_x$ such that $x^* = \arg\min_C g_j(x)$, then

$$n^* = \arg\min_D g_j(n), \tag{9}$$

is located within $\overline{B_{x^*}(r)} = \{x : \|x - x^*\|_2 \leq r\}$ with $r = \sqrt{2(g_j(n_{\text{init.}}) - g_j(x^*))/\lambda}$ for any given $j \in \{1, 2, 3\}$, where $\lambda$ is the smallest eigenvalue of the Hessian of $g_j(x^*)$ and $n_{\text{init.}} = \arg\min_E g_j(n)$ with $E = \{n \in \mathbb{N}^k : \sum n_i = \eta, \lfloor x_i^* \rfloor \leq n_i \leq \lceil x_i^* \rceil\}$.

Applying the result of Lemma 4.5, we propose Algorithm 1 which starts with the continuous solution, identifies a small set of candidate integer solutions, and then identifies the integer-optimal design within this smaller set.

**Algorithm 1** Integer-Optimal Design
***
**Input:** $x^*$ (the optimal continuous solution) and Hessian
    matrix of $g$ : $H_g(x^*)$
    **for** $i = 1, \dots, k-1$ **do**
        Define $T_i = \{n_i \in \mathbb{N} : \lfloor x_i^* \rfloor \le n_i \le \lceil x_i^* \rceil\}$
    **end for**
    Define $T = \{(n_1, \dots, n_{k-1}, n_k) : n_k = \eta - \sum_{i=1}^{k-1} n_i, \text{where } (n_1, \dots, n_{k-1}) \in T_1 \times \dots \times T_{k-1}\}$
    Select $n_{\text{init.}} = \arg\min_{n \in T} g(n)$
    Calculate the smallest eigenvalue $\lambda$ of $H_g(x^*)$
    Calculate radius $r = \sqrt{2(g(n_{\text{init.}}) - g(x^*))/\lambda}$
    **for** $i = 1, \dots, k-1$ **do**
        Define $S_i = \{n_i \in \mathbb{N} : \max(x_i^* - r, 1) \le n_i \le \min(x_i^* + r, N_i, \eta - k + 1)\}$
    **end for**
    Define $S = \{(n_1, \dots, n_{k-1}, n_k) : n_k = \eta - \sum_{i=1}^{k-1} n_i, \text{where } (n_1, \dots, n_{k-1}) \in S_1 \times \dots \times S_{k-1}\}$
    Select $n^* = \arg\min_{n \in S} g(n)$ by an exhaustive search.
**Output:** $n^*$
***

**Theorem 4.6** (Integer-Optimal Design). *Algorithm 1 outputs the integer-optimal design* (9).

*Remark* 4.7. Before entering the second for-loop in Algorithm 1, the practitioner may decide to check the optimality gap $[g(n_{\text{init.}}) - g(x^*)]/g(x^*)$. If the optimality gap is sufficiently small, it may be acceptable to adopt the suboptimal design $n_{\text{init.}}$. This is demonstrated in Section 5.4.

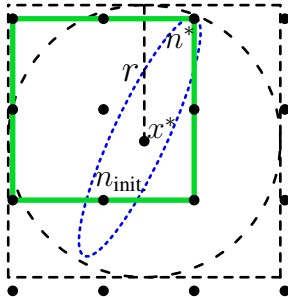

*Figure 2.* Illustration of Algorithm 1: level set (dotted blue), $\overline{B_{x^*}(r)}$ and its enclosing sphere and cube (dashed black), feasible region induced from $r$ (thick green), integer points (black)

Figure 2 illustrates the main idea behind Lemma 4.5 and Algorithm 1. Note that the dotted blue ellipse represents the level set of the variance objective $g$ evaluated at $n_{\text{init.}}$, the nearest integer design from the continuous optimum $x^*$. Starting by locating $n_{\text{init.}}$, our method then constructs a ball $\overline{B_{x^*}(r)}$ centered at $x^*$ with radius $r = \sqrt{2(g_j(n_{\text{init.}}) - g_j(x^*))/\lambda}$. This ball corresponds to the level set of a quadratic lower bound of $g$, and thus still contains the integer-optimal design. We then enclose the ball in a cube to simplify the search. The solid green cube

marks the final exhaustive search region, which includes all feasible candidates for the integer-optimal design.

*Remark* 4.8. Strong convexity of the variance objective plays a key role in identifying the optimal design. While an exhaustive grid search has $O\left(\binom{\eta-1}{k-1}\right)$ different combinations, scaling $O(\eta^{k-1})$ when $k$ is fixed, Algorithm 1 reduces $\eta$ to $2r$, giving a much lower complexity of $O\left((2r)^{k-1}\right)$. We show in Section 5 that $r$ is relatively small.

## 5. Simulation

We numerically illustrate our method through simulation studies. Section 5.1 compares compares variances between naive and DP-aware stratified sampling. Section 5.2 explores the interplay between the non-private and purely DP designs. Section 5.4 showcases the computational efficiency of our algorithm. The input of Algorithm 1, $x^*$, is obtained by package `nloptr` and `alabama` in R. All computations, including runtime measurements, were conducted on the Purdue Bell clusters using multiple cores. The source codes are available at https://github.com/garyUAchen/DP_OptimSurvey.

### 5.1. Suboptimality of Naive Stratified Subsampling

As shown in Table 1, stratified sampling under our DP framework requires a tailored design to minimize the variance objective. Both private mean estimation and private A-optimal estimation face variance inflation under specific privacy mechanisms.

In this simulation, there are 4 groups with population sizes $N = (7000, 8000, 9000, 10000)$ and variance $\sigma^2 = (0.08, 0.08^2, 0.08^3, 0.08^4)$ and a total sample size $\eta = 200$. We plot the variance ratio from a naive subsampling scheme to that of the integer-optimal design while varying $\epsilon$ from 0.01 to 100.

For the population mean case, Figure 3 illustrates that, under the Laplace mechanism, the naive subsampling variance can be up to 2.5 times larger than the optimal design variance within $1 < \epsilon < 10$. Under TuLap, the variance ratio can reach as high as 4. Note that DLap gives the same design for population mean.

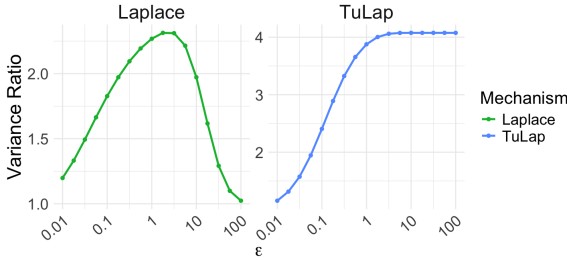

*Figure 3.* Variance Ratio on population mean

For A-optimal cases, illustrated in Figure 4, the naive sub-sampling variance under the Laplace mechanism can be up to 1.8 times larger than the optimal design variance within the range $1 < \epsilon < 10$. For the TuLap mechanism, a noticeable jump in the variance ratio occurs around $\epsilon = 10$, driven by the variance of the $\text{Unif}(0,1)$ component. Similarly, the DLap mechanism follows a trend comparable to TuLap but demonstrates the smallest variance ratio gap between the naive subsampling scheme and the optimal integer design. This behavior aligns with the population mean scenario, where the ratio remains constant at 1.

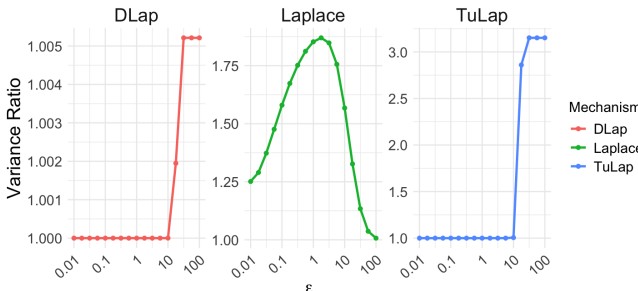

Figure 4. Variance Ratio for A-Optimal Design

The trend of the Laplace mechanism differs from those of DLap and TuLap because, in the latter two, strong convexity arises only from the data variance, while the variance induced from the Discrete Laplace mechanism is actually linear. In contrast, both the data and Laplace variance components are strongly convex in the Laplace mechanism. Intuitively, when both sources of randomness are strongly convex, each favors its own optimal design (for example, Neyman allocation for data variance and proportional allocation for purely Laplace variance in the population mean case). The interplay between these competing effects shapes the overall optimal design, as shown more clearly in Section 5.2.

## 5.2. Interplay between No-noise and Purely Laplace Noise Optimal Designs

While a closed-form solution for private mean estimation under the Laplace mechanism is unavailable, the optimal design tends to fall between the no-noise and pure-Laplace noise designs.

In our simulation, there are 3 groups with population sizes $N = (1000, 2000, 3000)$, $\sigma^2 = (0.08, 0.08^{1.5}, 0.08^2)$, $\epsilon = 1$ and a total sample size $\eta = 200$. We use the setting of the population mean with Laplace noise. Figure 5 demonstrates that the integer-optimal design largely interpolates between the no-noise and pure-noise designs. As privacy protection strengthens, the design shifts closer to the pure-noise configuration; conversely, with weaker privacy

protection the optimal design more closely aligns with the Neyman allocation.

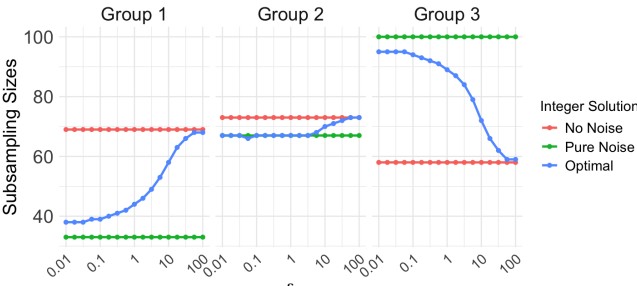

Figure 5. Optimal, no-noise, and pure-noise designs as $\epsilon$ varies

## 5.3. Sensitivity Analysis

The setting is the same as in Section 5.2, with additional consideration of misspecifying the sample variance $\hat{\sigma}^2$ relative to the true variance $\sigma^2$. We examine four cases where $\hat{\sigma}^2 = 3\sigma^2, 1.2\sigma^2, 0.9\sigma^2$, or $0.3\sigma^2$. As shown in Figure 6, the DP-optimal design is largely robust to misspecification within $\pm 20\%$. The convergence of curves near $\epsilon = 100$ occurs because, under weak privacy constraints, the DP-optimal design approaches Neyman allocation, which is proportional to $\sigma$, and thus to $\hat{\sigma}$ in this setting.

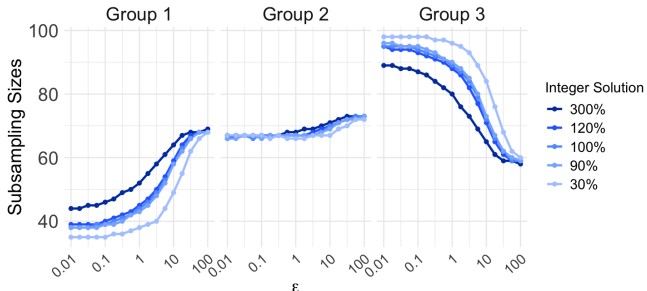

Figure 6. Sensitivity Analysis

## 5.4. Computational Efficiency of Algorithm 1

Our problem is formulated as a mixed-integer programming task, which lacks an off-the-shelf solution method, particularly in R. While exhaustive search becomes infeasible as either the total sample size $\eta$ or the number of groups $k$ increases, our algorithm proves to be relatively efficient for practical implementation. Although it struggles with scenarios involving large $k$, it remains highly efficient for substantial $\eta$ values—up to $100,000$ and more—when $k$ is kept at a reasonable scale.

In this simulation, there are 10 groups with $N = (N_1, N_2, \ldots, N_{10}) = (20000, 19000, \ldots, 11000)$ and $\sigma^2 = (\sigma_1^2, \sigma_2^2, \ldots, \sigma_{10}^2) = (0.08^{1.1}, 0.08^{1.2}, \ldots, 0.08^2)$

and $\epsilon = 1$. We measure the computation time for an exhaustive search and for our proposed algorithm as the total sample size $\eta$ increases from 30 to 48, using the population mean case with Laplace noise. Figure 7 demonstrates that the exhaustive search exhibits exponential growth in computation time, whereas Algorithm 1 effectively mitigates this growth. In fact, we see that Algorithm 1 can effectively find the optimal solution with sample sizes up to $10^5$ in less time than an exhaustive search takes for $\eta = 30$.

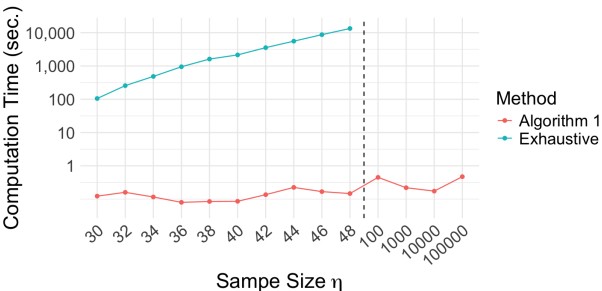

*Figure 7.* Computation time comparison

Now, we consider $k$ groups with $N = (N_1, N_2, \ldots, N_k) = (10000 + 1000 \cdot k, 10000 + 900 \cdot k, \ldots, 11000)$ and $\sigma^2 = (\sigma_1^2, \sigma_2^2, \ldots, \sigma_k^2) = (0.08^{1.1}, 0.08^{1.2}, \ldots, 0.08^{1+k/10})$ and $\epsilon = 1$. We implement our full algorithm from $k = 2$ to 12, locating the optimal design $n^*$, and that of the first part of our algorithm from $k = 14$ to 26, locating the nearest integer design $n_{\text{init.}}$.

If $r > 1$, then as $k$ increases the computation time exhibits exponential growth, which becomes large especially when $k \geq 14$. In practice, if $r > 1.5$ and $k$ is large, we recommend identifying the nearest integer design $n_{\text{init.}}$, corresponding to the first half of the algorithm. In this simulation, it maintains an optimality gap (the relative increase in the DP variance objective at $n_{\text{inti.}}$ compared to the continuous optimal $x^*$) of less than $10^{-4}$ from the continuous-optimal design $x^*$. Notably, locating the continuous-optimal design is independent of Algorithm 1 and requires less than 1 second for any $k$ in the simulation.

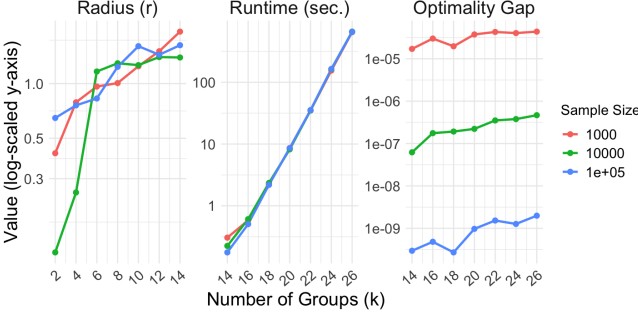

*Figure 8.* Curse of dimensionality in number of groups

## 6. Discussion

We proposed a new framework for integrating differential privacy (DP) into stratified sampling, aiming to achieve two main goals: minimizing the total variance and protecting individual privacy. A significant contribution of our work is the inclusion of privacy considerations in the data collection design. While traditional stratified sampling focuses on minimizing variance assuming non-private data, our approach takes into account the noise introduced by privacy mechanisms as well as the privacy amplification by subsampling, ensuring more reliable results under privacy constraints.

Our framework is fairly flexible, as it works with three common DP mechanisms: Laplace, Discrete Laplace, and Truncated-Uniform-Laplace. Furthermore, the strong convexity of the variance objective ensures that the optimization problem is well-defined, which leads to solid theoretical results. We also addressed the computational challenges of exhaustive search methods by developing an efficient algorithm for finding the optimal integer design.

However, there are some limitations to our framework. We assume prior knowledge of population variances across groups. In practice, a pilot study is commonly conducted, where a small portion of pilot samples are drawn from each group to estimate group sample variances. The fixed sample size constraint, while reasonable, may be too strict or may not account for more complex scenarios (e.g. chance constraints). Additionally, our algorithm may be inefficient when the search radius $r$ is large. Exploring alternative methods for finding the integer-optimal design could improve efficiency; for example `Gurobi`, `MOSEK`, and `Julia`'s `Pajarito` all have generic mixed-integer programming packages. Lastly, our analysis is limited to $\epsilon$-DP with the Laplace, DLap, and TuLap mechanisms and uses subsampling without replacement. Future work could extend our results to other privacy and subsampling frameworks, such as the Staircase mechanism Geng & Viswanath (2015), Gaussian mechanism, or general canonical noise distributions Awan & Vadhan (2023) as well as the $\rho$-zCDP (Bun & Steinke, 2016), $\mu$-GDP, $f$-DP frameworks (Dong et al., 2022) or subsampling with replacement (Balle et al., 2018).

Some directions for future work are as follows: Incorporating the double privacy amplification effect of subsampling and shuffling (Li et al., 2023a) could strengthen privacy guarantees but also introduces additional optimization challenges. Applying our framework in a fully central DP context, assuming a trusted curator, is another option and would yield a different variance objective. Finally, extending the framework to more complex sampling designs, such as multi-stage or adaptive sampling, would broaden its applicability. These directions would further solidify the role of DP-aware stratified sampling in privacy-preserving data collection.

## Impact Statement

This work enables accurate and privacy-aware data collection for private mean estimation by optimally designing stratified sampling schemes under differential privacy. It allows practitioners to conduct surveys in sensitive domains such as health, education, and public policy while obtaining reliable results backed by formal privacy guarantees.

## Acknowledgments

This work was supported in part by NSF grant no. SES 2150615 to Purdue University. The authors are grateful to the anonymous reviewers, whose valuable feedback helped to improve the presentation of this paper.

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

# A. Proofs and Technical Details

**Lemma A.1.** *For all $y > 0$,*

$$\left(\left(1 + \frac{1}{y}\right)\log(1+y) - 2\right)^2 - 1 + \log(1+y) > 0. \tag{10}$$

*Proof.* First, since we focus on the positive real line ($y > 0$), we can do change of variable by letting $y = e^x - 1$ with domain $x > 0$, then (10) becomes

$$\left(\frac{xe^x}{e^x - 1} - 2\right)^2 - 1 + x.$$

We can multiply through by $(e^x - 1)^2$ to give us the equivalent of problem of establishing that $h(x) > 0$, where

$$h(x) := (xe^x - 2e^x + 2)^2 + (x-1)(e^x - 1)^2.$$

To analyze $h(x)$, we compute its derivatives up to the third order:

$$\frac{dh(x)}{dx} = e^{2x}\left(2x^2 - 4x + 3\right) + 2e^x\left(x - 2\right) + 1,$$

$$\frac{d^2 h(x)}{dx^2} = 2e^x\left(e^x\left(2x^2 - 2x + 1\right) + x - 1\right),$$

$$\frac{d^3 h(x)}{dx^3} = 2xe^x(4xe^x + 1).$$

We begin by noting that $\frac{d^3 h(x)}{dx^3} > 0$ for all $x > 0$, which implies that $\frac{d^2 h(x)}{dx^2}$ is increasing. Since $\frac{d^2 h(x)}{dx^2}\big|_{x=0} = 0$, it follows that $\frac{d^2 h(x)}{dx^2} > 0$ for all $x > 0$. By the same argument, knowing that $\frac{dh(x)}{dx}\big|_{x=0} = 0$, we conclude that $\frac{dh}{dx} > 0$ for all $x > 0$. Finally, since $h(0) = 0$, we establish that $h(x) > 0$ for all $x > 0$. $\square$

**Theorem 4.1** (Strong Convexity). *Let $\alpha_i$, $N_i$, $\sigma_i$, and $\eta$ be given for all $i = 1, \ldots, k$. Then, the continuous relaxations of the variance objectives* (4), (5) *and* (6), *with $(n_1, \ldots, n_k)$ replaced by a continuous vector $x \in C_x := \{x \in \mathbb{R}^k : \sum x_i = \eta, 0 < x_i \le N_i, \forall i\}$, are strongly convex.*

*Proof.* Since we intend to prove that the variance objective is strongly convex on the reals, we substitute $x \in \mathbb{R}_+^k$ for $n$ in the proof for distinction. We will first prove that the variance objective is strongly convex as a function of $x$ (without the constraint $\sum x_i = \eta$) and then verify that they remain strongly convex under the constraint of $\sum_{i=1}^k x_i = \eta$.

Let $g_1(n) = $ (4). The Hessian matrix $H_f$ is a diagonal matrix with diagonal entry $(H_{g_1})_{ii}$ as follows:

$$\frac{\partial^2 g_1}{\partial x_i^2}(x_1, \ldots, x_k) = 2\alpha_i^2 \left[\frac{\sigma_i^2}{x_i^3} + \frac{(x_i + c_i)^2 \log^2(1 + \frac{c_i}{x_i}) - c_i(4x_i + 3c_i)\log(1 + \frac{c_i}{x_i}) + 3c_i^2}{x_i^3(x_i + c_i)^2 \log^4(1 + \frac{c_i}{x_i})}\right], \tag{11}$$

which is positive for $x_i \in (0, \eta)$ for all $i = 1, \ldots, k$, where $c_i = (\exp(\epsilon/\Delta f) - 1)N_i$. This is because the first term and the denominator of the second term of (11) are clearly positive for all $x_i \in (0, \eta)$ and hence it remains to check the numerator of the second term; by letting $y_i = \frac{c_i}{x_i}$ and completing the squares, we have

$$(x_i + c_i)^2 \log^2(1 + \frac{c_i}{x_i}) - c_i(4x_i + 3c_i)\log(1 + \frac{c_i}{x_i}) + 3c_i^2$$

$$= (\frac{c_i}{y_i} + c_i)^2 \log^2(1 + y_i) - c_i(4\frac{c_i}{y_i} + 3c_i)\log(1 + y_i) + 3c_i^2$$

$$= c_i^2 \left[(1 + \frac{1}{y_i})^2 \log^2(1 + y_i) - (3 + \frac{4}{y_i})\log(1 + y_i) + 3\right]$$

$$= c_i^2 \left[\left((1 + \frac{1}{y_i})\log(1 + y_i) - 2\right)^2 - 1 + \log(1 + y_i)\right]$$

$$> 0 \quad \text{(Lemma } A.1\text{)}.$$

Similarly, let $g_2(n) = (5)$ and $g_3(n) = (6)$. The Hessian matrix $H_{g_2}$ is a diagonal matrix with diagonal entry $(H_{g_2})_{ii} = 2\alpha_i^2 \left(\frac{\sigma_i^2}{x_i^3}\right)$ and the Hessian matrix $H_{g_3}$ is a diagonal matrix with diagonal entry $(H_{g_3})_{ii} = 2\alpha_i^2 \left(\frac{\sigma_i^2 + \frac{1}{12}}{x_i^3}\right)$, which are both positive for $x_i \in (0, \eta)$ for all $i = 1, \dots, k$.

Since $g_j$ is strongly convex with respect to $(x_1, \dots, x_{k-1}, x_k)$ for $j = 1, 2, 3$, it remains to show that they are strongly convex under the constraint $\eta = \sum_{i=1}^{k} x_i$. Let $P = \{(x_1, \dots, x_k) : \eta = \sum x_i\}$ and $a, b \in P$. For $t \in (0, 1)$, $g_j(ta + (1-t)b) < tg_j(a) + (1-t)g_j(b)$ because $a, b \in \text{dom}(g_j)$; meanwhile, $ta + (1-t)b \in P$ due to the convexity of the constrained subspace. Therefore, the strong convexity holds for an additional box constraint, i.e. $P' = \{(x_1, \dots, x_k) : \sum x_i = \eta, 0 < x_i < N_i\}$. $\qquad \square$

**Proposition 4.2** (Closed-Form Solutions for Population Mean). *If $M_i$ is DLap or TuLap with $\alpha_i = N_i$ and assume that $[\tau_i/(\sum \tau_i N_i)]\eta \leq 1$ for all $i$, the continuous solution of (5) and (6) under the constraint of $C_x$ have a closed form $x_i^* = [(\tau_i N_i)/(\sum \tau_i N_i)]\eta$, where $\tau_i^2 = \sigma_i^2$ for DLap and $\tau_i^2 = \sigma_i^2 + \frac{1}{12}$ for TuLap.*

*Proof.* Consider

$$\min \frac{1}{(\sum N_i)^2} \left\{ \sum_{i=1}^{k} \left[ \frac{N_i^2}{x_i} \tau_i^2 + 2 \frac{x_i + (\exp(\epsilon/\Delta f) - 1)N_i}{(\exp(\epsilon/\Delta f) - 1)^2} \right] \right\}$$

$$s.t. \sum x_i = \eta.$$

We can write its Lagrangian as

$$L(\mathrm{x}, \nu) = \frac{1}{(\sum N_i)^2} \left\{ \sum_{i=1}^{k} \left[ \frac{N_i^2}{x_i} \tau_i^2 + 2 \frac{x_i + (\exp(\epsilon/\Delta f) - 1)N_i}{(\exp(\epsilon/\Delta f) - 1)^2} \right] \right\} + \nu(\sum x_i - \eta).$$

The KKT condition implies that for all $1 \leq i \leq k$,

$$\begin{cases} \frac{\partial L}{\partial x_i} = \frac{1}{(\sum N_i)^2} \left[ \frac{-N_i^2}{x_i^2} \tau_i^2 + \frac{2}{(\exp(\epsilon/\Delta f) - 1)^2} \right] + \nu = 0 \\ \frac{\partial L}{\partial \nu} = \sum x_i - \eta = 0. \end{cases}$$

Then, $\nu(\sum N_i)^2 + \frac{2}{(\exp(\epsilon/\Delta f) - 1)^2} = \frac{N_i^2}{x_i^2} \tau_i^2$ implies $x_i \propto \tau_i N_i$, as the LHS of the equation is constant for all $i$. Therefore,

$$x_i^* = \frac{\tau_i N_i}{\sum \tau_i N_i} \eta,$$

where $\tau_i^2 = \sigma_i^2$ for Discrete Laplace and $\tau_i^2 = \sigma_i^2 + \frac{1}{12}$ for TuLap. $\qquad \square$

**Proposition 4.4** (Closed-Form Solution of Purely Laplace Variance for Population Mean). *Under the constraint of $C_x$, the purely DP variance from the Laplace mechanism (8) is minimized at $x_i^* = (N_i/\sum N_i)\eta$ for all $i$.*

*Proof.* We can express the problem as an equality-constrained minimization problem

$$\min \frac{\sum N_i \varphi(\frac{x_i}{N_i})}{(\sum N_i)^2}$$

$$s.t. \sum x_i = \eta,$$

where, with $c = \exp(\epsilon/\Delta f) - 1$, $\varphi(y) = \frac{2}{y} \log^{-2}\left(1 + \frac{c}{y}\right)$ is strongly convex w.r.t. $y$. Its Lagrangian is

$$L(x, \nu) = \frac{\sum N_i \varphi(\frac{x_i}{N_i})}{(\sum N_i)^2} + \nu(\sum x_i - \eta).$$

The KKT condition implies that for all $1 \leq i \leq k$

$$
\begin{cases}
\frac{\partial L}{\partial x_i} = \frac{\varphi'(\frac{x_i}{N_i})}{(\sum N_i)^2} + \nu = 0 \\
\frac{\partial L}{\partial \nu} = \sum x_i - \eta = 0.
\end{cases}
$$

Thus, $-\nu \sum N_i = \varphi'(\frac{x_1}{N_1}) = \cdots = \varphi'(\frac{x_k}{N_k})$. Note that $\varphi$ is strongly convex if and only if $\varphi'$ is strictly increasing, which implies $\frac{x_1}{N_1} = \frac{x_2}{N_2} = \cdots = \frac{x_k}{N_k}$ is a sufficient and necessary condition for the system of equations to hold. Now, since $\sum x_i = \eta$ and $x_i \propto N_i$ (i.e. $x_i = bN_i$ for some $b \in \mathbb{R}$), $\sum x_i = b \sum N_i = \eta$ and hence $b = \frac{\eta}{\sum N_i}$, which gives

$$
\begin{cases}
x_i^* = \frac{\eta}{\sum N_i} N_i \text{ for all } i, \\
\nu^* = \frac{-1}{(\sum N_i)^2} \varphi'(\frac{\eta}{\sum N_i}).
\end{cases}
$$

$\square$

*Lemma* 4.5 (Range to Search for Integer-Optimal Design). Let $g_1, g_2, g_3$ be the objective function from (4), (5), (6) respectively, and $C_x$ such that $x^* = \arg\min_C g_j(x)$, then

$$
n^* = \arg\min_D g_j(n), \tag{9}
$$

is located within $\overline{B_{x^*}(r)} = \{x : \|x - x^*\|_2 \leq r\}$ with $r = \sqrt{2(g_j(n_{\text{init.}}) - g_j(x^*))/\lambda}$ for any given $j \in \{1, 2, 3\}$, where $\lambda$ is the smallest eigenvalue of the Hessian of $g_j(x^*)$ and $n_{\text{init.}} = \arg\min_E g_j(n)$ with $E = \{n \in \mathbb{N}^k : \sum n_i = \eta, \lfloor x_i^* \rfloor \leq n_i \leq \lceil x_i^* \rceil\}$.

*Proof.* Let $L_f(c) = \{x : f(x) \leq c\}$ be the level set of $f$ at $c$. For any fixed $j \in \{1, 2, 3\}$, let

$$
l_j(x) := g_j(x^*) + \nabla g_j(x^*)^\top (x - x^*) + \frac{\lambda}{2} \|x - x^*\|^2.
$$

We see that, for any $c \in \mathbb{R}$, $L_{g_j}(c) \subseteq L_{l_j}(c)$. Thus, we plug $g_j(n_{\text{init.}})$ in and get

$$
n^* \in L_{g_j}(g_j(n_{\text{init.}})) \subseteq L_{l_j}(g_j(n_{\text{init.}})).
$$

Now, to calculate the radius of $L_{l_j}(g_j(n_{\text{init.}}))$, we look at the boundary of it, where $\{x : l_j(x) = g_j(n_{\text{init.}})\}$. Then, the set of boundary points satisfy

$$
g_j(n_{\text{init.}}) = g_j(x^*) + \nabla g_j(x^*)^\top (x - x^*) + \frac{\lambda}{2} \|x - x^*\|^2
$$

$$
= g_j(x^*) + \frac{\lambda}{2} \|x - x^*\|^2.
$$

The conclusion follows as we set the radius $r = \sqrt{2(g_j(n_{\text{init.}}) - g_j(x^*))/\lambda}$ such that $n^* \in \overline{B_{x^*}(r)}$. $\square$

