# OpenReview forum: "Optimal Survey Design for Private Mean Estimation"
_ICML.cc/2025/Conference — ICML 2025 poster_

### Official Review · Reviewer_o79u · 2025-03-11

**Overall Recommendation:** 4

**Summary:**

This paper studies how to estimate a population mean from surveys collected from different groups of people. Differential privacy is required at the level of each group At a high level, the mechanism randomly samples users from each group, who then send in their responses plus noise. The population mean is then a weighted average of the received responses. There are free parameters of the algorithm, namely the number of users to sample in each group, and depending on the variance estimate, these parameters can be optimized. Specifically, for three common types of noise distributions, the number of users for each group is a convex optimization problem, and can be selected using convex optimization. For special parameter choices, the paper derives closed-form solutions for this optimization problem, and in general, they show that the optimal solution can be found using exhaustive search in k dimensions (k is the number of groups). They demonstrate experimentally, the optimization procedure can reduce variance by a factor of 2-4, and the searching algorithm runs efficiently for up to moderate (<30) values of k.

**Claims And Evidence:**

Most of the claims in the paper are adequately supported by theorems and experimental evidence. There is one claim I feel needs more evidence, which is the local differential privacy guarantee.

**Essential References Not Discussed:**

I cannot think of essential references not already discussed.

**Experimental Designs Or Analyses:**

I did not check the experimental designs, but they seem reasonable.

**Methods And Evaluation Criteria:**

The methods and evaluation criteria are adequate.

**Other Comments Or Suggestions:**

None

**Other Strengths And Weaknesses:**

The algorithms in the paper are simple and easy to implement. This applies to both the estimation scheme, which is based off of simple local DP methods, and to the optimization algorithm, where the authors are very clear about how the various convex optimization solvers are being used.

A negative about the work is that it requires lots of prior knowledge to be known about the sample, namely their variances. It is somewhat common to make such assumptions in statistics, but it seems a bit more complicated in the privacy setting since it requires knowing something a priori about the private data.

**Questions For Authors:**

There is currently no quantitative explanation of the local differential privacy guarantee of the algorithm. I believe it should be a factor of $$N_i / n_i$$ higher than the central DP guarantee. This seems like it could be quite high; are there examples where the sampling probability is not too low (like >0.3 for example)? Do we gain anything statistically by taking a large fraction of samples?

When can we expect to have accurate estimates of variances in each group in a private setting? Can we use existing work to privately measure the population variance under DP?

**Relation To Broader Scientific Literature:**

This paper fits in both the differential privacy and statistics literature. It continues a trend towards adding differential privacy to fundamental statistical methods, including stratified sampling, the setting considered here.

**Theoretical Claims:**

I did not closely check the theoretical claims, but they seem reasonable.

---

> ### Author Rebuttal · Authors · 2025-03-27
>
> Thank you for recognizing that our work contributes to the growing trend of incorporating differential privacy into fundamental statistical methodologies, particularly in stratified sampling.
>
> - Regarding the need for prior knowledge of variances, we agree that this is a limitation. However, in practice, statisticians can often estimate variances using historical data or prior knowledge. Alternatively, a portion of the privacy budget could be allocated for variance estimation. We will conduct a sensitivity analysis through simulations to assess the impact of mild variance misspecifications, ranging from 10-20%.
>
> - **In Proposition 3.4**, the nominal privacy budget $M_i$ is given by
>   $$\log \left( \frac{\exp(\epsilon/\Delta f) - 1 + q_i}{q_i} \right)-DP$$
>   where $q_i = \frac{n_i}{N_i}$. We sincerely apologize for the omission of "-DP" in our submission, which may have led to confusion. We have corrected this in our revised version.
>
>   Indeed, when the subsampling rate $\frac{n_i}{N_i}$ is relatively small, the nominal budget can be quite large. Nevertheless, our primary goal is to ensure central DP while providing some additional protection against the data curator through the (weaker) local-DP guarantee.
>
>   There could certainly be settings where the sample size is a moderate proportion of the population size, and having a larger sample would improve the statistical estimation by reducing the variance.
>
> - Accurate variance estimation within each group can be feasible given historical data or prior knowledge. Alternatively, a portion of the privacy budget could be allocated to estimate population variances during an initial phase.

---

### Official Review · Reviewer_RCeZ · 2025-03-13

**Overall Recommendation:** 4

**Summary:**

The authors propose a DP stratified sampling scheme that can be optimized for various objectives such as population mean estimation or an A-optimal design. The main contributions are a general algorithm to solve the mixed-integer programming problem this creates, as well as closed-form solutions for important settings.

**Claims And Evidence:**

All theoretical results are accompanied by proofs in the appendix.

**Essential References Not Discussed:**

I am not aware of any essential references not discussed.

**Experimental Designs Or Analyses:**

I examined each of the experiments the authors present and did not notice any major issues. However, there are a few features of the figures that the authors might further analyze. For example, we see a strong effect for the Laplace Mechanism in Figure 2 where the variance ratio increases monotonically for epsilon < 1 and then decreases monotonically for epsilon > 1. A similar effect is observed in Figure 6. What is happening here?

**Methods And Evaluation Criteria:**

The methods and evaluation criteria make sense for the problem.

**Other Comments Or Suggestions:**

1. The figures are difficult to read in their current form; I ask the authors to please update the text size in the figures to match the remainder of the document.
2. I had trouble understanding Table 1 in Section 1 given what had been introduced so far at that point in the work. It wasn't until after reading to Section 5 that I was able to go back and understand the point the authors were making. I would suggest the authors either add more context initially or move Table 1 to later in the document.

**Other Strengths And Weaknesses:**

The authors identified a gap in the DP literature, successfully derived a solution, and provided a thoughtful experimental evaluation.

**Questions For Authors:**

I have no additional questions for the authors.

**Relation To Broader Scientific Literature:**

The key contribution of the paper is optimal survey design for private estimation, which (as far as I am aware) has not been previously explored in the DP literature.

**Theoretical Claims:**

I looked through the proofs in Appendix A, but did not examine them in great detail. I did not notice any issues.

---

> ### Author Rebuttal · Authors · 2025-03-27
>
> We are thankful that you acknowledged that we identified a gap in the DP literature regarding survey sampling and provided a successful solution along with a thoughtful experimental evaluation. We hope that our responses below will address your comments.
>
> - **(Experimental Design Comment)** Thank you for the question. Please recall that the DP variance objective consists of two components: data variance and DP randomness. When analyzing the variance objective under the Laplace mechanism, we found that both components are strongly convex, though proving the strong convexity of the DP-induced variance required significantly more effort. Initially, we assumed that the Discrete Laplace and TuLap mechanisms would exhibit similar behavior due to their comparable shapes. However, we later discovered that for these two mechanisms, strong convexity arises solely from the data variance, while the variance due to the Discrete Laplace mechanism itself is merely convex (specifically, linear). This key difference explains why the Laplace case differs from the others.
>
>   Intuitively, when both sources of randomness exhibit strong convexity, they each lead to their own optimal designs (Neyman allocation for data variance and proportional allocation for purely Laplace variance). The competition between these two effects determines the optimal design. However, this dynamic does not hold for the Discrete Laplace and TuLap mechanisms, as their variance objectives are not strongly convex by themselves. We will add these insights to the simulation section.
>
> - We will update the text size in the figures to match the font size in the main content to improve readability.
>
> - To clarify, the table compares the non-private solution (optimal design) evaluated with the DP variance against the DP-optimal solution evaluated with the DP variance, and the values are the ratio of these variances (higher is worse). We will provide additional context for Table 1 to ensure it can be understood by the reader in the introduction section.

---

### Official Review · Reviewer_kmJ8 · 2025-03-14

**Overall Recommendation:** 5

**Summary:**

This submission is about designing stratified sampling schemes for surveys conducted with differential privacy. In stratified sampling, groups may be surveyed at different rates and these per-group estimates then combined. This is a ubiquitous survey method.

The survey setting is an important one for statistical privacy, but there is not much work on it. The starting point of this paper is that is may be better to design the survey with privacy in mind, rather than independently combining "the best nonprivate survey" with "the best privacy mechanism." Notably, the choice of stratification interacts with the privacy guarantees from subsampling.

For a class of surveys and privacy mechanisms, this paper shows how the task of finding the lowest-variance private survey design can be solved optimally and efficiently.

**Claims And Evidence:**

Yes.

**Essential References Not Discussed:**

None that I am aware of.

**Experimental Designs Or Analyses:**

The simulations seem appropriate.

**Methods And Evaluation Criteria:**

Yes.

**Other Comments Or Suggestions:**

none.

**Other Strengths And Weaknesses:**

The submission is well-written and well-constructed. I found it very easy to follow.

The analysis here is limited: we only consider a few noise distributions and a certain family of variance objectives. The approach assumes exact access to quantities which will not usually be known (e.g., the true per-stratum variances). This is not a fully-formed solution, but I regard it as a large step toward it.

**Questions For Authors:**

none.

**Relation To Broader Scientific Literature:**

The submission grounds itself well in the existing literature and clearly identifies the gap it fills.

**Theoretical Claims:**

I did not check proofs carefully; the claims seem to make sense.

---

> ### Author Rebuttal · Authors · 2025-03-27
>
> Thank you for recognizing the contributions of our work. While we acknowledge that this research does not fully resolve all related problems, we are pleased that you see it as a significant step forward. Please let us know if you have any further questions.

---

> > ### Comment · Reviewer_kmJ8 · 2025-04-08
> >
> > I have no questions at this time.

---

### Official Review · Reviewer_TMRh · 2025-03-14

**Overall Recommendation:** 3

**Summary:**

This paper develops a stratified sampling scheme that minimizes variance while ensuring differential privacy (DP) under the Laplace, Discrete Laplace, and Truncated-Uniform-Laplace mechanisms. The key insight is that stratified sampling can amplify privacy guarantees, but optimal allocation of samples across strata must account for the effect of privacy noise. The authors formulate the problem as an optimization task, determining the optimal subsampling sizes to minimize variance while maintaining a fixed total sample size. They prove the strong convexity of the variance objective, derive closed-form continuous solutions for specific DP mechanisms, and propose an efficient algorithm for finding the optimal integer solution. The results demonstrate that ignoring DP effects can lead to significant variance inflation, and their method offers a principled way to balance privacy and accuracy in survey design.

**Claims And Evidence:**

See questions

**Essential References Not Discussed:**

No

**Experimental Designs Or Analyses:**

See questions

**Methods And Evaluation Criteria:**

See questions

**Other Comments Or Suggestions:**

NA

**Other Strengths And Weaknesses:**

- Since the analysis is limited to three specific DP mechanisms, could the author discuss the extension to consider alternative frameworks such as Gaussian DP or Rényi DP? Would it benefit the utility or computation results?

- For equation (1),  do we need the constraint $n_ i\leq N_ i$? That is, do we allow sub-sampling with replacement?

- This framework seems to assume that $\sigma_i$, the population variances, are known. Though in the discussion, the authors claimed that a pilot study is commonly conducted in practice, would the small portion of pilot samples further introduce a large variance in the objective function? How is it going to affect the accuracy of the solution?

**Questions For Authors:**

NA

**Relation To Broader Scientific Literature:**

NA

**Theoretical Claims:**

See questions

---

> ### Author Rebuttal · Authors · 2025-03-27
>
> Thank you for your questions. We are happy to address them below:
>
> - The key aspect of our setup that enables an efficient solution is the strong convexity property. When generalizing to other settings, it is important to verify that the resulting objective is still strongly convex, which may need to be done in a case-by-case manner; without strong convexity, our efficient algorithm is not applicable. Furthermore, we use the subsampling result of $\epsilon$-DP, which also exists in Rényi-DP and $f$-DP, but not in Gaussian-DP or zero-concentration DP. Regarding utility, the use of Gaussian noise could improve the utility of the final estimator as it has lighter tails than Laplace noise; however, the variances of both scale in the same manner as the privacy parameter is varied. Ultimately, these extensions are promising directions for future work, and we will include a paragraph in the discussion that includes these points.
>
> - You are correct that we do impose the constraint $n_i \leq N_i$, and we only consider subsampling without replacement. A possible extension of this work could also consider subsampling with replacement: for results on privacy amplification through subsampling with replacement, we refer to *"Privacy Amplification by Subsampling: Tight Analyses via Couplings and Divergences"* by B. Balle et al. As noted earlier, the key aspect to investigate in this extension is whether their variance objective is strongly convex, but this is left for future work. This extension will also be mentioned in the discussion section of the paper.
>
> - Yes, we agree and recognize this limitation of known variances. In practice, and as is widely done in survey sampling, the variances will be replaced by estimated variances or known bounds on the unknown variances. This is not ideal, but as other reviewers have also noted, this work begins the treatment of what appears to be a fundamental problem, and so assuming that variances are known seems like a reasonable step. Future work should address the issue of estimating variances, and especially the effect of any heavy tails that may result. Regarding accuracy, even if the variances are misspecified, the mean estimation remains unbiased, though the design may no longer be optimal. We will conduct a sensitivity analysis through simulations to compare the design performance under mild variance misspecifications of approximately 10-20%.
>
> Please let us know if our responses have addressed your questions or if you have any remaining concerns about the paper.
>
> We would appreciate any further clarification regarding your reasoning for the inclination towards rejection. We are particularly keen to understand your perspective, as other reviewers have acknowledged the novelty and potential impact of our work.

---

> > ### Comment · Reviewer_TMRh · 2025-04-02
> >
> > Thank you for your rebuttal and clarifications. I have updated the overall rating to 3. However, I still have a question regarding the constraint related to "subsampling without replacement." From my understanding, the constraint is currently defined as $\sum n_i = \eta, \ n_i \in \mathbb{N}$ in all relevant parts of the paper, including Equation (1) (Lines 329 left - 165 right), the definition of $D$ (Line 324), and Equation (9). However, it seems that the condition for "subsampling without replacement" is not explicitly included in these formulations.
> >
> > Specifically, from my understanding, the condition $n_i \leq N_i$ is a necessary condition for "subsampling without replacement." As such, the feasible set should be a subset of $\{n_i \in \mathbb{N} \ | \ \sum n_i = \eta, \ n_i \leq N_i\}$. Could the authors clarify how the "without replacement" condition is formulated in your setting?

---

> > > ### Author Response · Authors · 2025-04-03
> > >
> > > Thank you for raising your rating and for your question about including the constraint $n_i \leq N_i$. This was indeed a typo, and we have corrected it in the revised version. To be clear, all our calculations did include this constraint, even though our exposition omitted it in error.  As our work builds on the subsampling results from *Gaussian Differential Privacy* (Dong et al., 2022) and Ullman’s notes (2017), where $q_i = \frac{n_i}{N_i} \leq 1$, all our results are derived under the constraints of this feasible set.
> > >
> > > Please let us know if we have addressed all of your comments, or if there are any other questions or concerns that we can address.

---

### Decision · Program_Chairs · 2025-05-01

**Decision:**

Accept (poster)

**Comment:**

This paper introduces a principled framework for designing optimal stratified sampling schemes under differential privacy (DP) to estimate population means. The authors identify and address a key challenge: while stratified sampling is a classical technique to reduce variance, naïvely applying it under DP, without accounting for privacy amplification effects, can significantly degrade statistical efficiency. The paper proposes a variance-minimizing design tailored to the Laplace, Discrete Laplace, and Truncated-Uniform-Laplace mechanisms, and provides both theoretical analysis (including convexity guarantees) and efficient integer optimization algorithms for implementation.

This paper addresses a timely and foundational problem in the intersection of statistics and differential privacy, and does so with rigor, clarity, and practicality. It should be of interest to researchers in both theoretical DP and statistics.